# Biodistribution PET/CT Study of Hemoglobin-DFO-^89^Zr Complex in Healthy and Lung Tumor-Bearing Mice

**DOI:** 10.3390/ijms21144991

**Published:** 2020-07-15

**Authors:** Łukasz Kiraga, Gabriele Cerutti, Agata Braniewska, Damian Strzemecki, Zuzanna Sas, Alberto Boffi, Carmelinda Savino, Linda Celeste Montemiglio, Daniel Turnham, Gillian Seaton, Alessandra Bonamore, Richard Clarkson, Adam M. Dabkowski, Stephen J. Paisey, Bartłomiej Taciak, Paulina Kucharzewska, Tomasz P. Rygiel, Magdalena Król

**Affiliations:** 1Department of Cancer Biology, Institute of Biology, Warsaw University of Life Sciences, 02-787 Warsaw, Poland; lukkir@outlook.com (L.K.); bartek.taciak@gmail.com (B.T.); pb.kucharzewska@gmail.com (P.K.); 2Department of Biochemical Sciences “Alessandro Rossi Fanelli”, Sapienza University of Rome, 00-185 Rome, Italy; gabriele.cerutti@uniroma1.it (G.C.); alberto.boffi@uniroma1.it (A.B.); alessandra.bonamore@uniroma1.it (A.B.); 3Department of Immunology, Medical University of Warsaw, 02-097 Warsaw, Poland; agata.braniewska@gmail.com (A.B.); damian.strzemecki@gmail.com (D.S.); sas.zuza@gmail.com (Z.S.); trygiel@wum.edu.pl (T.P.R.); 4Institute of Molecular Biology and Pathology, National Research Council, 00-185 Rome, Italy; linda.savino@uniroma1.it (C.S.); lindac.montemiglio@uniroma1.it (L.C.M.); 5European Cancer Stem Cell Research Institute (ECSCRI), School of Biosciences, Haydn Ellis Building, Cardiff University, Cardiff CF24 4HQ, Wales, UK; TurnhamD@cardiff.ac.uk (D.T.); SeatonG1@cardiff.ac.uk (G.S.); clarksonr@cardiff.ac.uk (R.C.); 6Wales Research & Diagnostic PET Imaging Centre (PETIC), School of Medicine, Heath Park, Cardiff University, Cardiff CF14 4XN, Wales, UK; DabkowskiA@cardiff.ac.uk (A.M.D.); PaiseySJ@cardiff.ac.uk (S.J.P.)

**Keywords:** hemoglobin, protein drug carrier, biodistribution, tumor, immune cells, PET/CT, ^89^Zr, maleimide-deferoxamine

## Abstract

Proteins, as a major component of organisms, are considered the preferred biomaterials for drug delivery vehicles. Hemoglobin (Hb) has been recently rediscovered as a potential drug carrier, but its use for biomedical applications still lacks extensive investigation. To further explore the possibility of utilizing Hb as a potential tumor targeting drug carrier, we examined and compared the biodistribution of Hb in healthy and lung tumor-bearing mice, using for the first time ^89^Zr labelled Hb in a positron emission tomography (PET) measurement. Hb displays a very high conjugation yield in its fast and selective reaction with the maleimide-deferoxamine (DFO) bifunctional chelator. The high-resolution X-ray structure of the Hb-DFO complex demonstrated that cysteine β93 is the sole attachment moiety to the αβ-protomer of Hb. The Hb-DFO complex shows quantitative uptake of ^89^Zr in solution as determined by radiochromatography. Injection of 0.03 mg of Hb-DFO-^89^Zr complex in healthy mice indicates very high radioactivity in liver, followed by spleen and lungs, whereas a threefold increased dosage results in intensification of PET signal in kidneys and decreased signal in liver and spleen. No difference in biodistribution pattern is observed between naïve and tumor-bearing mice. Interestingly, the liver Hb uptake did not decrease upon clodronate-mediated macrophage depletion, indicating that other immune cells contribute to Hb clearance. This finding is of particular interest for rapidly developing clinical immunology and projects aiming to target, label or specifically deliver agents to immune cells.

## 1. Introduction

Hemoglobin has been recently rediscovered as a drug and oxygen carrier for biomedical applications due to its unique pharmacokinetics properties [1,2,3,4]. Free Hb released from erythrocytes in hemolytic processes is avidly scavenged by haptoglobin (Hp), capable of forming a tight Hp-Hb complex with almost femtomolar affinity in humans, one of the strongest protein–protein interactions observed so far [5]. Hp is able to bind Hb dimers predominantly interacting with the Hb α-chain subunits and overlapping with the interface between the two αβ dimers [6]. The Hp-Hb complex is then actively scavenged by macrophages through CD163 [7]. This scavenger receptor, selectively expressed by the monocytic cell lineage, especially in liver and spleen, is thought to be essential for Hb clearance in order to prevent the toxic effect of the heme molecule and to avoid kidney filtration with subsequent protein precipitation and tissue damage. Notably, Hb clearance mediated by liver is dose-dependent [2].

The natural capability of macrophages to internalize Hb might be exploited to transport and deliver different substances chemically linked to Hb to cancerous or infected cells. Indeed, macrophages and monocytes are continuously recruited into infected and inflamed tissues to counteract pathogenesis development [8,9]. However, there is evidence that macrophages, rather than being tumoricidal, can adopt a pro-tumoral phenotype. Clinical studies and experimental mouse models have demonstrated that macrophages may stimulate angiogenesis, enhance invasion and metastasis and act as immunosuppressants, preventing attack on tumor cells by T cells and natural killer (NK) cells [10]. The dual aspect of the macrophage function enabled the development of a double macrophage-based approach in anticancer therapy: (i) target and kill tumor-associated macrophages, or (ii) re-program their phenotype from a pro-tumoral M2 to an anti-tumoral M1, exploiting their plasticity and diversity to polarize their phenotype into different subsets in response to environmental cues [11]. In both approaches, Hb represents a promising protein-based carrier of macrophage-targeting drugs.

In this framework, we investigated the in vivo biodistribution of Hb, comparing healthy and lung tumor-bearing mice in order to validate its potential application as a drug carrier targeting tumor tissue. We labelled Hb with a bifunctional chelator, the maleimide-modified deferoxamine, loaded with the radionuclide ^89^Zr, and injected for subsequent imaging. This long-life radioisotope (with half-life of 3.27 days) was chosen to enable a sufficiently long in vivo biodistribution study. In order to track the complex, we used the positron emission tomography–computed tomography (PET/CT) imaging system. In this manuscript we demonstrate the feasibility of the method proving the Hb-DFO conjugation by X-ray crystallography and matrix-assisted laser desorption/ionization-time of flight mass spectrometry (MALDI-TOF MS), the quantitative ^89^Zr uptake by Hb-DFO using radiochromatographic methods and finally analyzing the biodistribution of the Hb-DFO-^89^Zr complex over time in naïve and lung tumor-bearing mice. To understand the role of macrophages in mediating Hb internalization in the liver we depleted liver macrophages by clodronate treatment, prior to monitoring of fluorophore-conjugated Hb by means of flow cytometry and In-Vivo MS FX PRO imaging. Surprisingly, in the absence of liver macrophages, we observed an increase in the Hb fluorescence signal from liver due to increased Hb uptake by other immune cells (CD45-positive leukocytes). To the best of our knowledge, this report is the first bio-imaging study describing Hb tracking by the PET/CT method.

## 2. Results

### 2.1. Formation of the Human Hb-DFO Conjugate Is Selective, Fast and Quantitative

The Hb-DFO synthesis reaction involves mixing Hb with DFO at neutral pH, followed by buffer exchange in order to eliminate the excess of DFO. Two DFO molecules per Hb tetramer were identified as covalently bound to the β chain of the Hb tetramer by MALDI-TOF MS analysis (Appendix A). The observed molecular weights obtained from the average mass of the doubly charged ions of the hemoglobin α and β chains are in agreement with their calculated values, respectively, 15,126.4 Da for the α chain and 15,867.2 Da for the β chain in their heme free form. In the case of Hb-DFO adduct, the peak of the beta chains shifted to 16,579.0 Da, thus corresponding to the addition of one molecule of maleimide-DFO (+711.5 Da). In both profiles, the smaller feature with a moiety with molecular weight 97 Da detected for both α and β globins (m/z 15225 for α globin; m/z 15965 for β globin) might be explained in terms of the addition of sulphuricorphosphoric acid present in the natural protein [12].

Subsequent X-ray diffraction studies demonstrated unambiguously that a single Hb-DFO adduct is formed, namely the maleimide-DFO reacted with cysteine residue at position β93, located in the α_1_β_2_ interface (Figure 1).Human Hb-DFO crystals diffracted at 1.55 Å resolution and were shown to belong to the P2_1_2_1_2_1_ space group with the unit cell containing the whole tetramer. Refinement of the model gave R_factor_ = 16.7% and R_free_ = 21.2% (Appendix A). The electron density clearly shows a ferric form of Hb with water bound to the heme iron, except for one of the α subunits that has the heme group in double conformation, one of them being in the oxygenated form (50% occupancy). In order to assess the effect of maleimide-DFO derivatization we superposed the Hb-DFO Cα atoms with those of the native human aquomet-hemoglobin (aquomet-Hb) in the R- [13] (3P5Q) and in the T-state [14] (1HGB). The root-mean-square deviation (RMSD) calculated using the Superpose program [15] were 0.44 for the R-state and 0.80 for the T-state, confirming that Hb-DFO is in the R-state and indicating that the presence of the DFO does not affect the overall quaternary R-like structure typical of ferric Hb. The map around the side chain of Cys93 of both β subunits clearly displays additional electron density corresponding to the maleimide moiety and the first six atoms of the DFO as a double conformer (Figure 1a). Indeed, the chelating long chain is unstructured in the absence of a divalent metal, and crystallographic data do not show specific interactions that would lock it in a fixed conformation.

### 2.2. DFO Slightly Affects the Orientation of Two Side Chains

The largest displacement with respect to native aquomet-Hb is observed for Cys93 and Asp94: Upon reaction with the maleimide-DFO these two residues adopt a different orientation, with the terminal part of their side chain pointing outward (Figure 1b). This is the only conformation of Cys93 compatible with the steric hindrance of the maleimide ring, which consequently displaces Asp94 out of the way of the long chelating chain of DFO. Thus, the accessibility of Cys93 not only allows the thiol group to act as a nucleophile toward the maleimide ring, but also exposes DFO chain to the aqueous environment where the chelating chain is available to bind metal ions in solution. Careful inspection of other amino acids on the β chains, that are displaced with respect to native aquomet-Hb (Glu43, Lys66 and Asp79), were shown to lay on the protein surface and are involved in crystal contacts (Appendix A, lower panel). For α chains there are no relevant differences in side chain positions except for His45, Lys60 and Arg92, involved in crystal contacts, and the notable exception of Trp14, which occupies the position that in native aquomet-Hb hosts a toluene molecule used as crystallization adjuvant (Appendix A, upper panel).

### 2.3. ^89^Zr Uptake by Hb-DFO Is Fast and Efficient

Hb-DFO bioconjugate was investigated for binding to ^89^Zr by radiometric methods. ^89^Zr uptake by Hb-DFO was carried out under similar conditions as described for uptake by antibodies [16], namely in 1 M oxalic acid adjusted to pH 7.0 by sodium carbonate. Quantitative ^89^Zr uptake was then estimated by radio-thin layer chromatography (radio-TLC) and radio-high performance liquid chromatography (radio-HPLC) (Figure 2). Data demonstrate that ^89^Zr was efficiently transferred from its water-soluble oxalate complex to the DFO chain moiety within 2 h incubation at 25 °C. Subsequent extensive washing through centrifugal concentration was sufficient to guarantee a radioactive load up to 78 MBq per mg of protein. Specific activity in our study was 4.87 GBq/µmol.

To assess the stability of Hb-DFO-^89^Zr, the solution of obtained complex was left overnight at 37 °C and then, radio-TLC was performed. The chromatogram shows a single peak reflecting the Hb-DFO-^89^Zr complex (Appendix A). Because ^89^Zr oxalate was not detected, the result indicates that the complex is stable under these conditions.

### 2.4. Dose-Dependent Biodistribution and Elimination of Hb-DFO-^89^Zr in Mice: No Preferential Internalization Observed by Damaged Tissues

In order to validate the use of Hb-DFO-^89^Zr tracer for monitoring the body distribution of Hb in the context of routine in vivo imaging, we investigated its biodistribution in healthy mice by means of PET/CT. In this framework, we also used Hb-DFO-^89^Zr to monitor Hb biodistribution in a mouse model of lung cancer, exploiting Hb as a potential bio-tool to selectively transport diagnostic agents to cancer cells.

Hb-DFO-^89^Zr complex was administered intravenously at two different doses, namely 0.03 and 0.09 mg, to isoflurane-anesthetized naïve and lung tumor-bearing mice. PET imaging, followed by CT, was performed for each mouse and repeated at specific time-points post injection. Emission data, collected for 30 or 15 min at a spatial resolution of 0.4 mm are shown in Figure 3a–d. The development of mammary cancer lung metastases followed intravenous injection of 4T1 cells on the basis of the isogenic transplantation, and it was confirmed ex vivo at the end of the experiment (Figure 3e,f).

Topographic regions of liver, spleen, kidneys and lungs were quantitatively analyzed and normalized to the total radioactivity of the whole body. Overall, the observed biodistribution indicate that Hb-DFO-^89^Zr complex was rapidly scavenged by liver and excreted with urine regardless the administered dose (Figure 3 and Figure 4). When the Hb-DFO-^89^Zr complex was administered at 0.03 mg, it predominantly migrated to liver but also was present in smaller amounts in spleen, lungs and kidneys (Figure 3a,b and Figure 4a). When the Hb-DFO-^89^Zr complex was administered at three-fold higher dose (0.09 mg), the highest radioactivity of all organs was found in kidneys (Figure 3c,d and Figure 4b). Furthermore, spleen, lungs and liver exhibited lower radioactivity related to the injected dose (% ID/mL) than in the case of 0.03 mg administration (Figure 4a,b). The increased Hb-DFO-^89^Zr distribution to the tumor-bearing lungs compared to the lungs of healthy mice was not observed (Figure 4b). Gradual, dose-dependent loss of the radiotracer was observed as shown in Figure 4c.

### 2.5. Hb Is Not Scavenged by the Liver Macrophages in a Mouse Model

High accumulation of Hb in liver can be explained by the fact that liver macrophages—Kupfer–Browicz cells—constitute a natural clearance pathway, which reduces Hb-Hp circulation time and toxicity [11]. Liver macrophages express CD163, also termed “hemoglobin scavenger receptor”, which is known to scavenge Hb-Hp complexes [7] and Hb alone but with lower affinity [17]. In order to assess the role of macrophages in Hb uptake by the liver, we depleted liver macrophages by the intravenous administration of liposomal clodronate [18]. After 24 h, we injected Hb conjugated with the AlexaFluor 750 fluorophore (AF750). Using flow cytometry, we proved that clodronate treatment was very efficient and removed almost all liver macrophages assessed as F4/80^+^cells (Figure 5a). Surprisingly, depletion of liver macrophages resulted in increased total fluorescence of Hb (Hb-AF750) in total liver immune cells expressing pan-leukocyte marker CD45 (Figure 5b). Accordingly, ex vivo measurement of total Hb-fluorescence in major organs showed that macrophage depletion led to accumulation of Hb signal in liver. There were no significant differences in Hb-fluorescence in other measured organs, namely spleen, kidneys or lungs (Figure 5c). The above results suggest increased liver Hb uptake after macrophage depletion.

## 3. Discussion

Positron emission tomography with computer tomography scanning is a unique tool for drug development [19]. As a non-invasive imaging method, PET has an unrivalled sensitivity when monitoring the biodistribution of various compounds when radiolabeled with short living positron-emitting radioisotopes. This is of particular relevance in the precise assessment of targeted therapy or biodistribution of biological compounds [20]. Combining a reasonably long half-life radioisotope for protein biodistribution with excellent PET image quality and quantitation, ^89^Zr (half-life 3.3 days) represents an appealing choice for imaging tissue distribution of biologic therapeutics (e.g., antibodies [21,22,23] or proteins [24,25]).

In this work ^89^Zr was used as a radiolabel to monitor the in vivo biodistribution of human Hb injected in mouse models by PET/CT. Hb has recently moved to the center stage of biomedicine as a potential carrier of drugs [3,4,26,27]; it can be readily obtained either directly from patients or as a highly purified biopharmaceutical protein, it is biocompatible, non-antigenic, it undergoes naturally controlled degradation processes, it is amenable to genetic or chemical modifications and it shows unique pharmacokinetic properties [3,4,28,29]. We employed chemically modified Hb with maleimide-deferoxamine as a chelating agent for ^89^Zr. The structure of Hb was only marginally and locally affected by DFO conjugation, since its canonical tetrameric state is retained, as confirmed by X-ray crystallography and MALDI spectrometry.

The PET imaging of Hb-DFO-^89^Zr complex revealed its major uptake by liver, followed by spleen and then lungs. The radioactivity related to the injected dose of these organs slightly decreased when a higher dose of Hb-DFO-^89^Zr (0.09 mg) was injected as compared with organs obtained from mice injected with a lower dose of Hb-DFO-^89^Zr (0.03 mg). Injection of the higher dose of Hb-DFO-^89^Zr also resulted in intense visualization of kidneys where most of Hb-DFO-^89^Zr is accumulated. This observation indicates that, after injection of 0.09 mg of Hb-DFO-^89^Zr, the serum Hp binding capacity was exceeded and Hb circulating in the free, unbound state, is accumulated in kidneys for clearance, typically resulting in hemoglobinuria [30]. Despite the high dose, injected Hb does not accumulate in tumor-bearing lungs of mice (as confirmed ex vivo). This observation shows that injected Hb is not preferentially transported to damaged tissues in mouse models of lung tumor, indicating some limitations in employing the 4T1 lung-tumor mouse model as a system to study plain Hb properties in vivo as carrier for drugs and diagnostic agents in cancer research.

Hb has a high affinity for CD163 after binding with Hp that could potentially make it an attractive tool for macrophage targeting [31]. However, our study showed that intravenously injected Hb in the complex with DFO-^89^Zr or AF750 fluorophore accumulates at significantly higher levels in livers of mice deprived of macrophages by the clodronate treatment as compared to livers of control animals. Furthermore, we showed that Hb is predominantly transported to leukocytes (CD45-positive cells) residing in the liver, indicating that Hb is taken up by other leukocytes than CD163^+^ macrophages in mice, supporting findings that Hb clearance differs in mice and humans [32]. This finding should be supported by further studies.

The present data indicates the feasibility of human Hb based carriers for delivery and labeling of murine liver immune cells. These findings are particularly interesting for rapidly developing clinical immunology and projects aiming to specifically deliver agents to immune cells, e.g., residing in liver cancers. There are many limitations to conventional approaches for liver cancer therapy. Various nanocarriers tested for liver cancer have shown importance for solving these problems. Development of a variety of nanotechnology platforms (e.g., theranostic, where an isotope is used), therefore has considerable promise as the next generation of medicine that enables early detection of disease, simultaneous monitoring and treatment and targeted therapy with minimal toxicity to diagnose and treat liver issues [33]. The findings of this manuscript are therefore important for the field of drug delivery research.

## 4. Materials and Methods

### 4.1. Synthesis of Hb-DFO

Human Hb from a healthy donor was purified as described by Perutz [34]. Hb modification was performed by reacting 20 molar excess per heme of maleimide-DFO (Macrocyclicsinc, Plano, TX, USA) with oxy-Hb in PBS buffer pH 7.4. Reaction was carried out on a 10 mg/mL Hb solution (2 mL) mixed with 12 mg/mL maleimide-DFO (0.2 mL) at 25 °C for 2 h. The Hb-DFO solution was then washed 5 times (1:10 dilution) in a VIVASPIN 30 MWCO (Sartorius AG, Goettingen, Germany) centrifugal concentrator. Final solution was 6.2 mg/mL in 2 mL PBS.

### 4.2. MALDI-TOF Analysis of Hb-DFO

Protein sample (40 µL) were desalted on a C8 Empore Disk (3M, Minneapolis, MN, USA) home-made stage tip and resuspended in 3 µL 1% formic acid. A volume of 1 µL was spotted on a MALDI sample plate and allowed to air dry. Recrystallized sinapinic acid (SA matrix from Thermo Fisher Scientific, Waltham, MA, USA) was prepared at a concentration of 5 mg/mL in 50:50 acetonitrile/water (0.1% Formic Acid) and spotted directly prior to insertion into the mass spectrometer. MALDI mass spectra were acquired on a 4800 MALDI-TOF/TOF mass spectrometer (Applied Biosystems, Foster City, CA, USA) equipped with a nitrogen laser operated at 336 nm. Acquisitions were performed in linear mode averaging 2500 laser shots in a random, uniform pattern. Ions were accelerated with a 20 kV pulse, with a delayed extraction period of 860 ns. Spectra were generated by averaging between 500 and 2000 laser pulses in a mass range from 4 kDa to 50 kDa. Laser intensity was set to optimize the signal-to-noise ratio and the resolution of mass peaks of the analyte. All spectra were externally calibrated and processed via Data Explorer software (version 4.7, Applied Biosystems, Waltham, MA, USA).

### 4.3. Crystallization of Hb-DFO

Human Hb-DFO was crystallized using the batch method described by Perutz [31]. In a total volume of 60 µL, crystals were grown at 20 °C in a solution containing 10 mg/mL Hb-DFO, 2.4 M (NH_4_)_2_SO_4_, 0.23 M (NH_4_)_2_HPO_4_, 0.07 M (NH_4_)H_2_PO_4_, pH 5.8. First crystals required six months to reach an appreciable size; control of nucleation was obtained through the microseeding technique [35]. A few crystals were transferred in 100 µL of crystallization cocktail and then vortexed for 1 min: 1 µL of the resulting seed stock was added to the crystallization batch, allowing crystals to grow in two days. Crystals were harvested in cryoloops, transferred in mother liquor containing 18% glycerol and flash-frozen in liquid nitrogen for data collection.

### 4.4. X-ray Data Collection, Structure Determination and Refinement

X-ray diffraction data of Hb-DFO were collected at 100 K at ELETTRA (Trieste, Italy), beamline XRD1, using a PILATUS detector. Data was indexed, scaled and integrated using the XDS package (built 20180808, MPI, Heidelberg, Germany) [36]. Molecular replacement was carried out using MOLREP ver. 11.0 [37] from the CCP4 suite (ver. 7.0, STFC Rutherford Appleton Laboratory, Harwell, UK) [38]. As search model we used the structure of ferric R-state aquomet-Hb at 2.0 Å resolution [13] (PDB 3P5Q). Iterative automated and manual structure refinement was carried out using REFMAC5 ver. 5.5 [39] and COOT ver. 0.8 [40], both implemented in the CCP4 suite [38]. Manual adjustment was performed with the F_o_-F_c_ map contoured at 1σ and the 2F_o_-F_c_ map at 3σ. Data processing and structure refinement statistics are shown in Appendix A. Figures were produced with Chimera (ver. 1.13, RBVI, San Francisco, CA, USA) [41]. The atomic coordinates and the structure factors have been deposited in the Protein Data Bank with accession code 6HK2.

### 4.5. Radiosynthesis of ^89^Zr Oxalate

^89^Zr was produced in house by the ^89^Y(p,n) ^89^Zr reaction via an adaptation of the methods of Walther et al. [42] and Dabkowskiet al. [43]. Briefly, a disk of natural abundance ^89^Y foil (300 µm thick, Goodfellow, Huntingdon, UK) in a custom made aluminium holder was loaded into a COSTIS Solid Target System (STS) fitted to an IBA Cyclone (18/9) cyclotron (Brussels, Belgium) equipped with a 400 µm thick niobium beam degrader. The disk was irradiated for 4 h with a beam of energy of 40 µA. The irradiated disk was left in the cyclotron for 12 h to allow any short lived ^89m^Zr to decay to ^89^Zr before removal for purification (activity 1.5-2 GBq). The disk was then dissolved in 2 M HCl (Fisher Scientific, Loughborough, UK) with stirring and heat and the ^89^Zr was isolated by flowing over a hydroxymate functionalized ion exchange resin column (freshly prepared in house for each separation). The column was rinsed with 2 M HCl and water to remove ^89^Y before the ^89^Zr was eluted with 1 M oxalic acid (Fisher Scientific, Loughborough, UK) and collected in 3 fractions of 1 mL. The most concentrated fraction contained 800–1000 MBq, which equates to a specific activity of 8.9–11.1 MBq/mg of oxalic acid (800–1000 MBq/mmol).

### 4.6. Radiolabelling of Hb-DFO

^89^Zr solution in 1 M oxalic acid was adjusted to pH 7.0 by adding 0.5 M sodium carbonate (Fisher Scientific, Loughborough, UK). A volume of 400 µL of the neutralized solution (radioactivity—35.2 MBq) was added to 57 µL of Hb-DFO (6.2 mg/mL). The resultant 457 µL reaction mixture was incubated at 25 °C for 2 h in a thermomixer apparatus under shaking. The solution was then centrifuged in VIVASPIN 30 MWCO concentrator (2000 rcf, 10 min) in order to remove excess salt in solution. Flow-through was discarded; the concentrated protein fraction was diluted in 450 µL of PBS and centrifuged using the same VIVASPIN column. Washing in PBS (Fisher Scientific, Loughborough, UK) and centrifugation was repeated one more time. Final precipitate was dissolved in 125 µL PBS. Radioactivity of the solution was 12.72 MBq. UV-vis absorption measurement (DeNovix DS-11, Wilmington, DE, USA) indicated that concentration of protein in solution was 1.3 mg/mL (total mass of Hb-DFO-^89^Zr in solution 0.162 mg, specific activity 78.3 MBq/mg or 4.87 GBq/µmol). Afterwards, radio-TLC and radio-HPLC were performed in order to evaluate free zirconium (unchelated by DFO).

### 4.7. Radio-TLC

A volume of 5 µL of reaction mixture was spotted onto a 12 × 1 cm strip of salicylic acid impregnated glass fibre ITLC paper (Agilent Technologies, Santa Clara, CA, USA), strips were developed for 5–10 min (solvent front 110 mm) with 50 mM DTPA solution adjusted to pH 7.4 using NaOH. The developed strips were read with a Canberra iSCAN radio-TLC reader controlled by Laura version 4.14 (scan speed 0.6 mm/s) (Lablogic Systems Ltd., Sheffield, UK).

### 4.8. Radio-HPLC

Radio-HPLC was carried out as described by Knight et al. [16] using an Agilent 1200 series machine (Agilent, Santa Clara, CA, USA) equipped with a refractive index detector, a single wavelength UV detector set to 254 nm, a Gamma-Ram 4 radiodetector (Lablogic Systems Ltd., Sheffield, UK) and a Superdex 200 10/300 GL size exclusion column (GE Healthcare LifeSciences, Marlborough, MA, USA). Phosphate buffer eluent was prepared as described by Vosjan et al. [22]. A volume of 1 µL of reaction mixture diluted in PBS up to 40 µL was injected and fractions of 20 µL were collected at a flow of 0.5 mL/min for 1 h, column temperature 35 °C.

### 4.9. Transplantation of 4T1 Cells to BALB/c Mice

The 4T1 cells (breast cancer cell line obtained from mouse mammary gland carcinoma) obtained from ATCC (no. CRL-2539) were cultured under optimal conditions: Gibco RPMI 1640 medium (Thermo Fisher, Waltham, MA, USA) containing l-glutamine and phenol red, supplemented with 10% (*v*/*v*) FBS and penicillin-streptomycin (50 U/mL) in standard culture conditions in an atmosphere of 5% CO_2_ and 95% humidified air at 37 °C; 4T1 cells were detached using trypsin, viability was checked using 0.4% trypan blue solution (Sigma-Aldrich, St. Louis, MO, USA) and samples containing 0.5 × 10^6^ viable cells suspended in 100 µL of PBS were prepared. For the experiments, BALB/c mice were housed in the conventional, non-SPF facility of The Wales Research and Diagnostic Positron Emission Tomography Imaging Centre (PETIC). All animal procedures were carried out in accordance with guidelines set by the UK Home Office in compliance with the Animals Scientific Procedures Act of 1986 under the UK HO project license number: 3003433. All animal work was approved by the University of Cardiff, School of Biosciences Research Ethics Committee; project identification codes: GM130/66 (1/3/2011) and GM130/671 (10/1/2015).

BALB/c mice were obtained from Harlan Laboratories (Indianapolis, IN, USA). Animals were acquired at six to eight weeks of age and maintained in individually ventilated cages (Allentown Inc., Allentown, NJ, USA) with a 12 h day/night cycle. Mice received a Teklad global 19% protein extruded rodent diet (Harlan Laboratories) and water ad libitum.

To establish a mouse model of triple negative breast cancer metastasis, which mimics lung tumor lesions [44], mice were anesthetized with 3–3.5% isoflurane (Piramal Critical Care LTD, West Drayton, UK) delivered through a nose cone, placed on a heating pad and intravenously injected (tail vein) using insulin syringes with 4T1 cells (0.5 × 10^6^). After 9 days, the mice were euthanized and the lungs were collected and fixed in Bouin’s solution (Sigma-Aldrich, St. Louis, MO, USA) in order to visualize lung tumors and to confirm transplantation efficiency.

### 4.10. Hb-DFO-89Zr PET/CT Imaging of BALB/c Mice

For the experiments, BALB/c mice (females, 8–12 weeks old) were housed in the conventional, non-SPF facility of The Wales Research and Diagnostic Positron Emission Tomography Imaging Centre (PETIC). All experimental protocols were approved by the Local Ethical Committee. Mice were selected, each mouse was anesthetized with 3–3.5% isoflurane/100% O_2_ (Piramal Critical Care LTD, West Drayton, UK) delivered through a nose cone and placed on the heating pad. Samples for injections were prepared. A volume corresponding to ~2 or ~6 MBq of radioactive substance was collected from previously prepared Hb-DFO-^89^Zr solution and, once diluted to 50 µL with PBS, it was intravenously injected (tail vein) using insulin syringes; each sample contained ~0.03 (*n* = 2) or ~0.09 mg of protein (*n* = 5). Afterwards, mice were recovered from inhalation anesthesia. Before PET/CT imaging, each mouse was IP injected with 100 µL of iopamidol (Niopam 300, Bracco, Milan, Italy) (CT contrast agent, 10 min before acquisition) and anesthetized with 3–3.5% isoflurane (Piramal Critical Care LTD, West Drayton, UK) delivered through a nose cone and placed on the heating pad. Mice were placed in the prone position in the scanner with the use of the Mediso Multicell 3 mouse animal bed (Mediso, Budapest, Hungary). The 1.5–2%/100% O_2_ isoflurane anesthesia was maintained through the entire image acquisition. The PET imaging, followed by CT, was performed using PET/CT Preclinical Imaging System (nanoScan122S PET/CT Mediso, Budapest, Hungary). Respiration was monitored with pressure pad connected to differential pressure transducers for low-range pressure monitoring during the entire PET/CT examination. PET scans were performed twice for each mouse, 1 h and 30 h after injection. Emission data were collected for 30 or 15 min. Spatial resolution of PET measurements was 0.4 mm and energy lower/upper limit was 400/600. The CT scan parameters were set as follows: Tube voltage was 50 kVp, tube current was 1 mA, exposure time 300 ms and maximum number of projections was 400. The resulting PET and CT metadata was reconstructed and the DICOM files of the PET and CT images were fused using PMOD software, version 3.806, module PFUS (PMOD Technologies LLC, Zurich, Switzerland). Reconstructed resolution of CT was 0.25 mm. The image of each injected and scanned mouse was subjected to a quantitative analysis and the results obtained were later analyzed statistically. For quantitative analysis module PBAS was used (PMOD Technologies LLC, Zurich, Switzerland). An organ shape on fused PET-CT image was contoured on each slide consisting part of this organ. Obtained VOI (volume of interest) was quantitatively analyzed using PBAS module of PMOD and the value consisting kBq per 1 mL of VOI was taken. Collected values were decay-corrected for T1/2 of ^89^Zr = 78.41 h related to time elapsed from intravenous administration to the end of PET signal acquisition. Finally, %ID/mL values (per cent of injected dose per 1 mL of VOI) were calculated.

### 4.11. In-Vivo MS FX PRO Imaging and Flow Cytometry

For the experiment, BALB/c mice (females, 10 weeks old) were housed in the conventional, non-SPF Animal Facility of Medical University of Warsaw. All experimental protocols were approved by the Second Local Ethics Committee for Animal Experimentation in Warsaw (license number: WAW2/138/2019) and all methods were performed in accordance with the relevant guidelines and regulations. Six mice were injected intraperitoneally with 5 mL/kg of Clodronate Liposomes (Liposoma B.V., Amsterdam, The Netherlands) and 5 untreated mice served as a control group. After 24 h mice were injected with 0.2 mg of human Hb (Sigma-Aldrich, St. Louis, MO, USA) labeled with Alexa Fluor 750 (Thermo Fisher Scientific, Waltham, MA, USA). After 24 h mice were sacrificed and livers, spleens, kidneys and lungs were collected. Organs were imaged using In-Vivo MS FX PRO (Carestream, Rochester, NY, USA). After imaging, livers were cut into small pieces, digested using Collagenase type IV (600 U; Sigma-Aldrich, St. Louis, MO, USA) and DNAse (400U, Sigma-Aldrich, St. Louis, MO, USA) for 1 h at 37 °C. Then tissue was dissociated using gentleMACS™ Dissociator (Miltenyi Biotec, Bergisch Gladbach, Germany) and cell suspension was passed through 100 µm Falcon Cell Strainer (Corning Life Sciences, Tewksburry, MA, USA). Next, erythrocytes were lysed using Gibco ACK Lysing Buffer (Thermo Fisher, Waltham, MA, USA). Then, cells were stained with Zombie Aqua™ Fixable Viability kit (BioLegend, San Diego, CA, USA) for 20 min at room temperature, blocked with 5% normal rat serum (STEMCELL Technologies, Vancouver, Canada) for 15 min on ice and then incubated with following antibodies: Anti-CD11b-Alexa Fluor 488 (macrophage marker) (eBioscience, San Diego, CA, USA), anti-F4/80-APC (macrophage marker) (eBioscience, eBioscience, San Diego, CA, USA) and anti-CD45.2-PE-Cyanine7 (general leukocyte marker) (Tonbo Biosciences, San Diego, CA, USA) for 20 min on ice. Samples were analyzed using flow cytometry FACSCanto II system (BD Biosciences, Franklin Lakes, NJ, USA) and FlowJo software (ver. 10.5, FlowJo LLC, Ashland, OR, USA).

### 4.12. Statistical Analysis

The statistical analysis was conducted using Prism version 5.00 software (GraphPad Software, San Diego, CA, USA). The one-way ANOVA and t-test were applied. The *p*-value < 0.05 was regarded as significant, whereas the *p*-value < 0.01 and < 0.0001 was regarded as highly significant. The data is expressed as means +/− S.D.

## 5. Conclusions

In conclusion, we developed a new method of radiolabeling Hb that allows its pharmacokinetics to be followed in vivo in different physio/pathological conditions. Furthermore, we found a way to enhance the liver uptake of Hb by the pre-treatment of mice with liposomal clodronate.

## Figures and Tables

**Figure 1 ijms-21-04991-f001:**
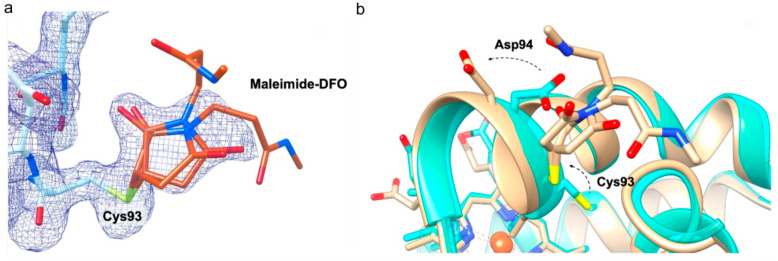
Local conformational changes observed for Cys93 and Asp94 upon reaction with maleimide-DFO in β chains. (**a**) Close up view on β1 Cys93 covalently bound to maleimide-DFO. The electron density map of the maleimide ring bound to Cys93 is shown and contoured at 1σ. In both β chains the maleimide ring and the first six atoms of DFO were identified in double conformations, 50% occupancy of the first six atoms of DFO and high B-factors do not allow electron density to be visible when contoured at 1σ. Carbon atoms are shown in orange for DFO, in white for Hb; nitrogen atoms are in blue, oxygen in red, sulfur in yellow (**b**) Secondary structure superposition of the native aquomet-Hb structure (transparent turquoise, PDB 3P5Q) with the Hb-DFO structure (opaque tan). The covalent bond between Cys93 and the maleimide-DFO induces a re-orientation of the side chains of Cys93 and Asp94, without affecting the main chain and the helical geometry of the F helix. Both residues move in the same direction, minimizing steric hindrance and exposing DFO to the solvent environment. Secondary structures and carbon atoms are shown in turquoise for native aquomet-Hb, in tan for Hb-DFO; nitrogen atoms are in blue, oxygen in red, sulfur in yellow, iron in orange.

**Figure 2 ijms-21-04991-f002:**
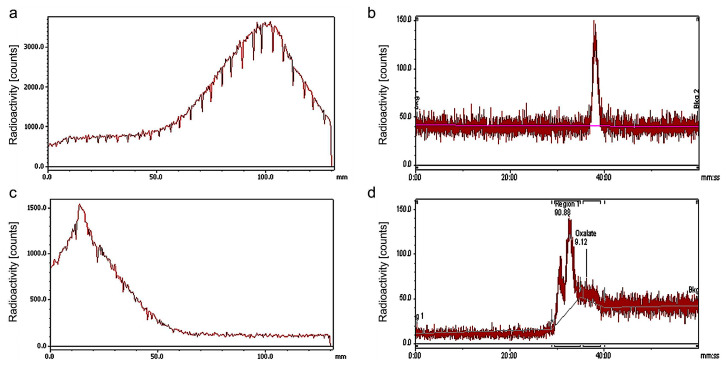
^89^Zr uptake by Hb-DFO: Radiochromatography analysis. Radio-TLC (**a**) and radio-HPLC (**b**) chromatograms of control ^89^Zr oxalate are reported; (**c**) and (**d**) show the radio-TLC and radio-HPLC chromatograms of Hb-DFO-^89^Zr, respectively. Radio-TLC analysis does not allow detection of zirconium oxalate signal, while in more sensitive radio-HPLC measurement only a low peak was observed. As the resolution of radio-HPLC chromatogram is higher than radio-TLC, two peaks for Hb-DFO-^89^Zr are observed, that indicates the presence of Hb aggregates in the product (reflected by the initial, lower peak).

**Figure 3 ijms-21-04991-f003:**
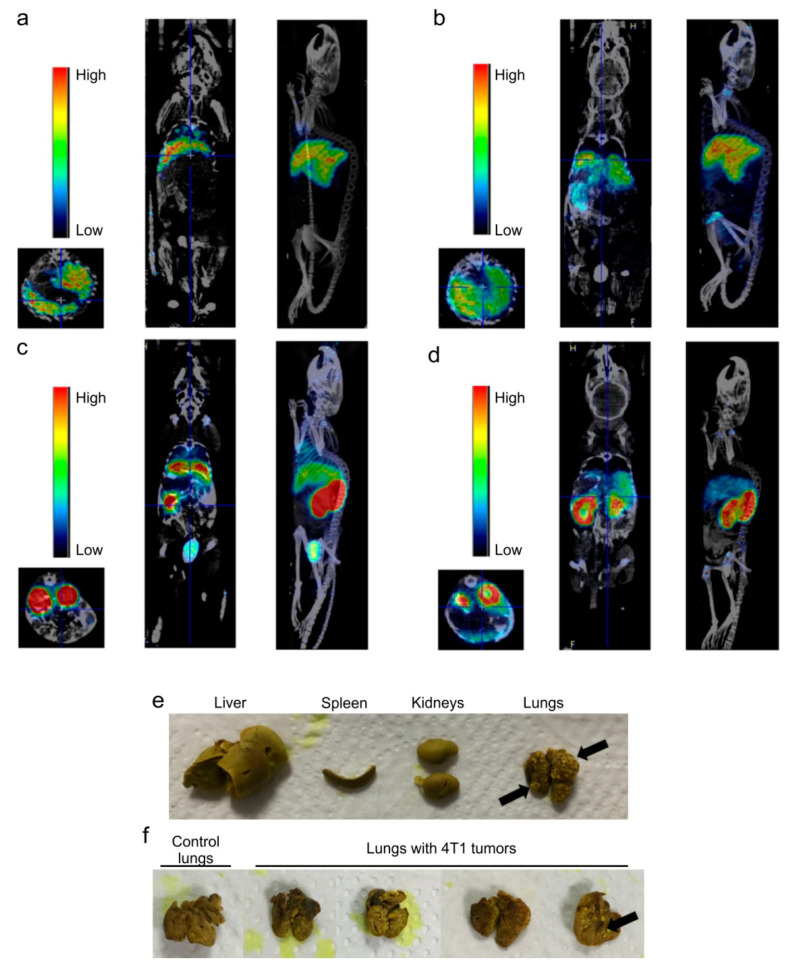
The biodistribution of Hb-DFO-^89^Zr complex in vivo. (**a**,**b**) Transverse plane, coronal plane and maximum intensity projection (MIP) of representative fused PET and CT images after intravenous injection of 0.03 mg of Hb-DFO-^89^Zr into healthy mouse; (**a**) shows signal detected 1 h after injection; high intensity signal from liver and low intensity signal from lungs; (**b**) shows signal detected 30 h after injection; high intensity signal from liver, reduction of signal from lungs. (**c**,**d**) Transverse plane, coronal plane and MIP of representative fused PET and CT images after intravenous injection of 0.09 mg of Hb-DFO-^89^Zr into tumor-bearing mouse; (**c**) shows signal detected 1 h after injection; high intensity signal was detected from kidneys and liver, low signal from lungs; (**d**) shows signal detected 39 h after injection; high intensity signal from kidneys, reduction of signal from liver and lungs. Radiotracer is eliminated and excreted with urine; color bars reflect radioactivity per volume. (**e**) Representative macroscopic images of liver, spleen, kidneys and lungs excised from mice that were inoculated with 4T1 cells nine days prior to organ collection. (**f**) Representative macroscopic images of lungs obtained from control mice and lung-tumor-bearing mice. Arrows indicate tumor nodules.

**Figure 4 ijms-21-04991-f004:**
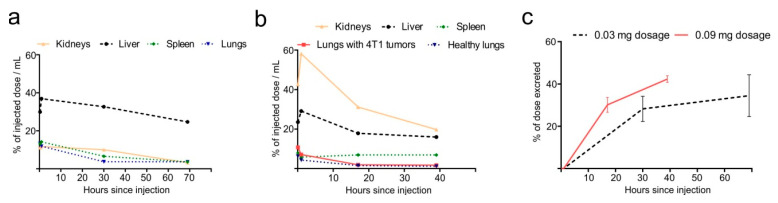
Quantitative analysis of positron emission tomography–computed tomography (PET/CT) images. (**a**) Mean % injected dose (ID)/mL for liver, spleen, kidneys and lungs calculated after imaging at different time-points (20 min, 1 h, 30 h and 69 h) after administration of 0.03 mg dose of Hb-DFO-^89^Zr. (**b**) Mean % ID/mL for liver, spleen, kidneys, lungs with 4T1 tumors and healthy lungs calculated after imaging at different time-points (5 min, 1 h, 17 h and 39 h) after administration of 0.09 mg dose of Hb-DFO-^89^Zr; measurements of radioactivity were decay-corrected. (**c**) Percentage of eliminated radiotracer at different time points following injection: 20 min, 1 h, 30 h, 69 h for 0.03 mg dose and 5 min, 1 h, 17 h, 39 h for 0.09 mg dose.

**Figure 5 ijms-21-04991-f005:**
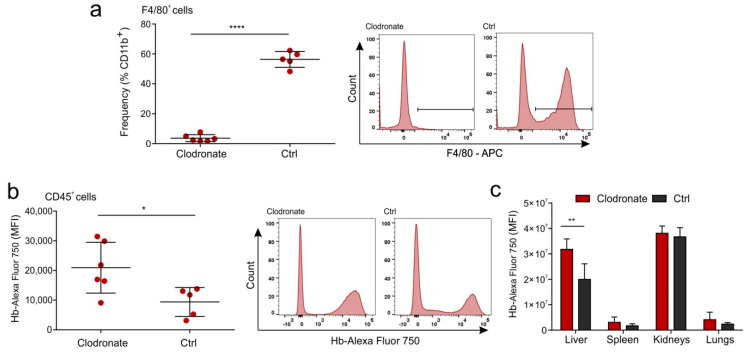
Hb is not scavenged by liver macrophages in a mouse model. Mice were depleted from macrophages by intravenous injection of liposomal clodronate 24 h before Hb administration. (**a**,**b**) Flow cytometry analysis of cells isolated from livers. (**a**) Frequency of F4/80^+^ cells in livers (shown as percentage of CD11b^+^ cells) from mice treated with liposomal clodronate or PBS. Graph (left) and representative histograms (right). (**b**) Mean fluorescence (MFI) of Hb-Alexa Fluor 750 in liver CD45^+^ cells. Graph (left) and representative histograms (right). (**c**) Mean fluorescence (MFI) in the Hb-channel of entire organs collected 24 h after intravenous injection of 0.2 mg of Hb-Alexa Fluor 750 measured using MS FX PRO imaging system. Significance was calculated with *t*-test. * *p* < 0.05; ** *p* < 0.01; **** *p* < 0.0001.

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
