# Peer review of "Biodistribution PET/CT Study of Hemoglobin-DFO-^89^Zr Complex in Healthy and Lung Tumor-Bearing Mice"

_ijms, 2020, doi:10.3390/ijms21144991_

Round 1
Reviewer 1 Report
The authors have improved the quality of the manuscript.
Reviewer 2 Report
I review the added comments and met my concerns.
This is good for me with new tracer and interesting data.
This manuscript is a resubmission of an earlier submission. The following is a list of the peer review reports and author responses from that submission.
Round 1
Reviewer 1 Report
New tracer in animal model, good new data.
I saw abstract, methods, results and discussion, but I did not see the conclusion.
This needs to be added.
Reviewer 2 Report
Recommendation: It appears that publication in any form would be premature at this time.
Comments: The experiments were not carefully planned, and additional number of mice are necessary for quantification and statistical analyses. Therefore, I don’t think this paper is suitable for publication.
- It is not clear why the authors decided to radiolabel hemaglobulin (MW 65 g/mol) with a long-live radioisotope.
- The manuscript is missing MALDI analyses on the Hb-DFO.
- The number of mice used in experiments is not enough to account for biological variability or to provide statistical analyses. Additionally, biodistributions studies are necessary to determine differences in healthy vs. tumor-bearing mice.
- The quantity of Hb-DFO-89Zr injected in healthy mice is different from the one reported for tumor-bearing mice. What is the specific activity of the radioconjugate?
- Are the images shown in Fig. 3 after calibration?
- What is the significance of increasing Hb accumulation in the liver?
- What is the stability of Hb-DFO-89Zr?
Reviewer 3 Report
In this article the authors described a Hb chelated DFO, combined with 89Zr , to image murine breast-cancer implant to lungs by PET. The data shows that no significant biodistribution difference between healthy and tumor bearing mice. Thus the paper is a negative result report. Overall the manuscript is well prepared and written. However the significance of the content and interest to readers are lacking. The unexpected findings were not thoroughly explained or discussed. I have the following critics:
- Major: The rationale for hemoglobin tracking by PET was not well explained. In the introduction section the authors declared that 'to the best of knowledge, this is the first study describing Hb tracking by PET/CT method'. Here in this manuscript, Hb was only a tool to applied macrophage targeting drugs. The resolution of PET imaging technique has intrinsic limitations in the order of mm. It sounded as if the technique is important to reveal biophysics of Hb (or oxidative metabolism, for example Raichle's 15O CMRO2 studies in the 80s) and is not appropriate .
- It was not clear what the specific activity was in section 4.5-4.6
- Fig 2a. Please comment on the periodic spikes in the TLC.
- Major : Figure 3. No unit had been provided in the figures. It was not clear how normalization ('after decay corrected') was done. The readers do not understand whether the unit would be SUV (mean or max) or radioactivity per volume.
- Figure 3S must be promoted and embedded with the Fig 3 or other figures to make clear demonstration of successful plantation of breast tumor into the lungs.
- Method section- the authors mentioned repeatedly isoflurane with provider named a hospital without a vendor name. Please correct it. In addition, they must clearly state whether the mice were imaged with 100% oxygen (which is a common practice in hospital labs) or less concentrated oxygen (20% or so). This affects interpretation of data as hemoglobin concentration may saturate to 100% in blood.
- The authors mentioned surprising results that Hb signal increase without liver macrophages. The authors need to discuss the implication for this , for example liver cancers.
- Line 224 'moved to the center stage of bioimaging'. The statement cannot be found in any papers in 4,25-27.